# Optogenetic control of *Bacillus subtilis* gene expression

Sebastian M. Castillo-Hair [ID] [1], Elliot A. Baerman[2], Masaya Fujita[3], Oleg A. Igoshin [ID] [1,2,4] & Jeffrey J. Tabor [ID] [1,2]

The Gram-positive bacterium *Bacillus subtilis* exhibits complex spatial and temporal gene expression signals. Although optogenetic tools are ideal for studying such processes, none has been engineered for this organism. Here, we port a cyanobacterial light sensor pathway comprising the green/red photoreversible two-component system CcaSR, two metabolic enzymes for production of the chromophore phycocyanobilin (PCB), and an output promoter to control transcription of a gene of interest into *B. subtilis*. Following an initial non-functional design, we optimize expression of pathway genes, enhance PCB production via a translational fusion of the biosynthetic enzymes, engineer a strong chimeric output promoter, and increase dynamic range with a miniaturized photosensor kinase. Our final design exhibits over 70-fold activation and rapid response dynamics, making it well-suited to studying a wide range of gene regulatory processes. In addition, the synthetic biology methods we develop to port this pathway should make *B. subtilis* easier to engineer in the future.

[1] Department of Bioengineering, Rice University, 6100 Main St., Houston, TX 77005, USA. [2] Department of Biosciences, Rice University, 6100 Main St., Houston, TX 77005, USA. [3] Department of Biology and Biochemistry, University of Houston, 4800 Calhoun Rd., Houston, TX 77004, USA. [4] Center for Theoretical Biophysics, Rice University, 6100 Main St., Houston, TX 77005, USA. Correspondence and requests for materials should be addressed to J.J.T. (email: jeff.tabor@rice.edu)

*B*acillus *subtilis* is a model organism for studying how time-varying (dynamic), heterogeneous, and spatially-coordinated gene expression signals control single- and multicellular behaviors[1–4]. For instance, in response to short-term energy stress (i.e., glucose/phosphate limitation), transcription of the general stress response regulon is activated in a pulsatile manner with a frequency proportional to the stress intensity[5]. In contrast, environmental (e.g., osmotic) stress induces a single transcriptional pulse of this same regulon with an amplitude proportional to the rate of onset of the stress[6]. In a separate pathway, persistent starvation induces the master regulator of sporulation via a series of pulses of increasing amplitude[7–9]. Though all cells exhibit these pulses, only a subset go on to produce spores, possibly due to heterogeneity in levels of the master regulator[10]. Furthermore, prior to sporulation commitment, an excitable and noisy genetic circuit drives a small fraction of cells to transiently differentiate into a state competent for DNA uptake[11–13]. From a spatial perspective, coordinated gene expression patterns can be observed in biofilms of undomesticated *B. subtilis*, where subpopulations of motile, matrix-producing, and sporulating cells localize to different regions[14]. Furthermore, in these biofilms, metabolism and growth are synchronized across cells via potassium-mediated action potentials that radiate outwards from the center[15,16].

Despite the richness of these regulatory dynamics, the underlying genetic circuits are typically studied using static and spatially homogenous genetic perturbations, including genetic knockouts[17] or the use of chemically-inducible promoters to express genes of interest at various steady state levels[18,19]. The inability to program artificial gene regulatory signals with precise temporal and spatial features in *B. subtilis* has limited understanding of even the most intensely-studied pathways. In contrast, engineered light-sensing two-component systems (TCS) have enabled exceptional control of gene expression dynamics in *E. coli*[20], even at the single cell level[21]. Similar tools that enable precision control of *B. subtilis* gene expression are needed[22].

The *Synechocystis* PCC6803 TCS CcaSR is a green light-activated/red light de-activated transcriptional regulatory pathway[23] (Fig. 1). CcaSR comprises the cyanobacteriochrome (CBCR)-family sensor kinase (SK) CcaS, and the OmpR/PhoB-family response regulator (RR) CcaR. CcaS contains a putative N-terminal transmembrane helix, followed by a cGMP phosphodiesterase/adenylyl cyclase/FhlA (GAF) domain, two Per-ARNT-Sim (PAS) domains of unknown function, and a histidine kinase (HK) domain. CcaS senses light via the linear tetrapyrrole chromophore phycocyanobilin (PCB), a prosthetic group covalently bound to the GAF domain. PCB is synthesized in two steps: heme oxygenase 1 (encoded by *ho1*) oxidizes heme to biliverdin IXα (BV), and phycocyanobilin:ferredoxin oxidoreductase (encoded by *pcyA*) reduces BV to PCB (Fig. 1a)[24,25]. Holo-CcaS (hereafter CcaS) is produced in a green light (535 nm)-sensitive ground state with low autokinase activity. Absorption of a green photon switches CcaS to a red light (672 nm)-sensitive active state with high autokinase activity. Active CcaS transfers a phosphoryl group from ATP to a conserved HK histidine residue (Fig. 1b), and subsequently to a conserved aspartate residue on CcaR (Fig. 1c). Phosphorylation activates CcaR, which subsequently induces transcription from the $P_{cpcG2}$ output promoter (Fig. 1c, d). Red light exposure reverts active CcaS to the ground state and de-activates $P_{cpcG2}$ transcription, likely through CcaS-mediated dephosphorylation of CcaR[26] (Fig. 1b).

We and others have repurposed the CcaSR system as an optogenetic tool, and utilized it to achieve exceptional quantitative, spatial, and temporal control of gene expression in *E. coli*. In the original study, we cloned the unmodified *ccaSR* genomic locus into an *E. coli* plasmid, which we co-transformed with a PCB

production plasmid encoding a synthetic *ho1-pcyA* operon[27]. However, this v1.0 system exhibits leaky output in red light and is activated less than 5-fold by green. In a follow-up study, we decreased leakiness and increased dynamic range to over 100-fold by systematically optimizing expression of *ccaS*, *ccaR*, and the *ho1-pcyA* operon, and truncating $P_{cpcG2}$ to remove an unwanted transcriptional start site (resulting in $P_{cpcG2-172}$)[28]. Sode and coworkers later constructed several miniaturized CcaS variants lacking the PAS domains, and demonstrated that two of these proteins result in lower $P_{cpcG2}$ output in red light and similar or greater $P_{cpcG2}$ output in green[29]. We introduced the corresponding CcaS PAS deletions in the context of our optimized (v2.0) system, resulting in an *E. coli* CcaSR v3.0 system with very low leakiness and nearly 600-fold dynamic range[30]. Various versions of the CcaSR system have been used alone and in combination with additional optogenetic tools with different wavelength specificities to achieve precise spatial[27,31] and temporal[20,32,33] control of the expression of one or multiple genes, including at the single-cell level[21]. In one of the studies, we programmed linear ramps and sine waves of expression of a transcriptional repressor in order to characterize the input/output (I/O) dynamics of a widely-used synthetic gene circuit[20].

Here, we combine lessons learned in our previous work with several novel synthetic biology approaches to port CcaSR into *B. subtilis*. Our initial design, which is based on *E. coli* CcaSR v2.0, does not respond to light. We utilize fluorescent protein fusions to reveal that *ho1*, *pcyA*, and *ccaS* are poorly expressed. Recoding of the initial ORF sequences and several modifications of the gene expression cassettes substantially improve expression. Despite these optimizations, we find that PCB levels remain low. Inspired by previous work on enzyme fusion[34] and scaffolding[35], we engineer a *ho1-pcyA* translational fusion, which results in high PCB levels. Next, we demonstrate that $P_{cpcG2-172}$, which is derived from *Synechocystis* PCC6803, is weak relative to other *B. subtilis* promoters. To increase transcriptional output, we chimerize $P_{cpcG2-172}$ with a strong constitutive *B. subtilis* promoter. Then, we increase CcaSR dynamic range by screening the best-performing miniaturized CcaS variants in the context of our system. Finally, we characterize the steady state and dynamic I/O of the optimized system, named *B. subtilis* CcaSR v1.0, to demonstrate that it should enable characterization of a wide range of *B. subtilis* gene regulatory processes. The principles elucidated during this debugging and optimization process should be of great utility to future *B. subtilis* synthetic biology applications, and more generally, to any situation where porting genetic circuits between bacterial species is required.

## Results

**B. subtilis CcaSR v0.1 does not respond to light**. Our first implementation, named CcaSR v0.1, comprised three independent modules integrated into distinct genomic loci: the PCB production module (PPM v0.1, Fig. 1a), light-sensing module (LSM v0.1, Fig. 1b), and transcriptional output module (TOM v0.1, Fig. 1c). This design was based on the insights obtained from our previously engineered *E. coli* CcaSR circuit, where dynamic range increased with higher expression of *ho1* and *pcyA* and was optimal at intermediate *ccaS* and *ccaR* expression levels[28]. Thus, we engineered a synthetic *ho1-pcyA* operon under control of the strong constitutive *B. subtilis* promoter $P_{rpsD}$[36] and separate copies of the synthetic ribosome binding site (RBS) MF001 (see "Methods" section) in PPM v0.1. Additionally, we expressed *ccaS* from the xylose-inducible promoter $P_{xylA}$ and its associated RBS[18] and *ccaR* from the IPTG-inducible promoter $P_{hy-spank}$[19] and synthetic RBS MF002 (see "Methods" section) in LSM v0.1 and TOM v0.1, respectively. Finally, to measure the output of the

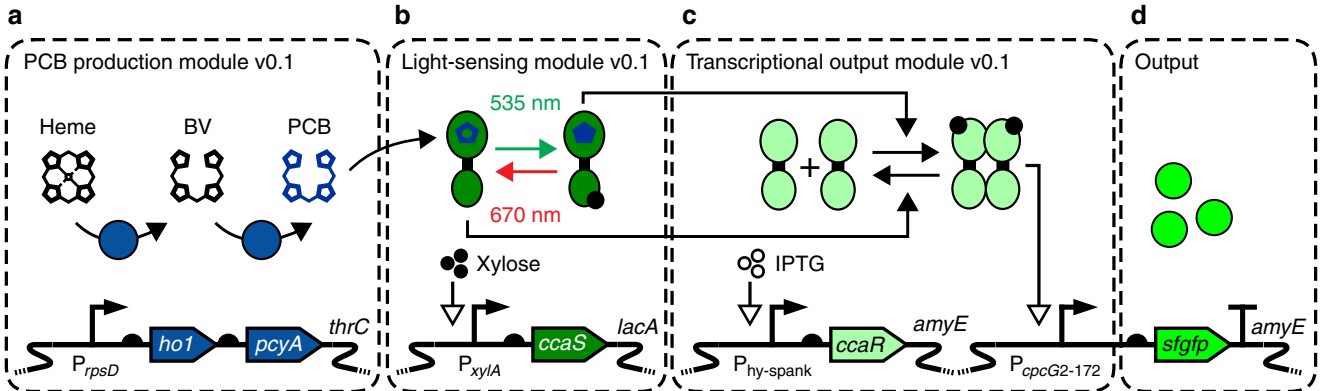

**Fig. 1** *B. subtilis* CcaSR v0.1 device schematic. The first implementation of *B. subtilis* CcaSR, comprised of **a** PCB production module v0.1, **b** light-sensing module v0.1, and **c** transcriptional output module v0.1. **d** System activity is measured by an *sfgfp* reporter. Genetic diagrams are shown at the bottom, and proteins, prosthetic groups, and expected interactions are represented at the top

system, we encoded superfolder green fluorescent protein (*sfgfp*) under $P_{cpcG2\text{-}172}$ and RBS MF001.

To characterize *B. subtilis* CcaSR v0.1, we induced expression of *ccaS* and *ccaR* by adding different combinations of xylose and IPTG concentrations under both red and green light, resulting in a total of 96 different conditions. Then, we assayed the output of the system by measuring the resulting sfGFP fluorescence in calibrated Molecules of Equivalent Fluorescein (MEFL) units via flow cytometry (see "Methods" section). We observed that even in the absence of xylose, sfGFP levels increased up to 40-fold in response to IPTG addition (Supplementary Fig. 1B, C), likely due to CcaR phosphorylation from a small molecule donor such as acetyl phosphate[37], while reporter activity was not detectable with a control TOM where the DNA binding domain of CcaR was removed (Supplementary Fig. 2). These results indicated that CcaR was properly expressed and capable of binding to and activating transcription from $P_{cpcG2\text{-}172}$ in *B. subtilis*. However, *B. subtilis* CcaSR v0.1 did not show any meaningful response to light under any of the conditions tested (Supplementary Fig. 1D). Furthermore, in contrast to *E. coli* CcaSR v2.0[28], addition of the *ccaS* inducer had little effect on transcription from $P_{cpcG2\text{-}172}$ (Supplementary Fig. 1B, C). Taken together, these results suggested that PCB was poorly produced, *ccaS* was poorly expressed, and/or CcaS was non-functional.

**Debugging and optimizing PCB production.** Cph1(Y176H) is an engineered *Synechocystis* PCC6803 phytochrome protein that becomes weakly red fluorescent upon PCB-binding[38,39]. We set out to utilize this protein to examine PCB levels produced from PPM v0.1 (Fig. 1a). To ensure that we could detect the fluorescent signal, we expressed *cph1(Y176H)* from our recent LacI-T7 promoter system, which results in very strong IPTG-inducible protein expression[40]. As a positive control, we added purified PCB (see "Methods" section) to *B. subtilis* cells overexpressing *cph1(Y176H)* with saturating IPTG. Surprisingly, the bacteria were very weakly red fluorescent (15.7 ± 1.9 Molecules of Equivalent Allophycocyanin, MEAP) (Supplementary Fig. 3A, B) (see "Methods" section). We suspected that *cph1(Y176H)* was poorly expressed. To examine this possibility, we constructed a *cph1(Y176H)-sfgfp* translational fusion and measured green fluorescence as before. Indeed, saturating IPTG yielded only 113 ± 27 MEFL of green fluorescence (Supplementary Fig. 3C, D) compared to 432,000 ± 20,000 MEFL when *sfgfp* is expressed directly from the same LacI-T7 system[40].

Highly expressed bacterial genes tend to exhibit little mRNA secondary structure near the RBS[41–44]. Computational analysis revealed that our synthetic RBS MF001 formed stable secondary structure with the initial *cph1(Y176H)* ORF sequence (Supplementary Fig. 3E). To increase expression, we replaced several of the first 15 codons with synonymous versions (resulting in *cph1(Y176H)**) predicted to reduce the problematic structure (Supplementary Fig. 3F). Indeed, this recoding process yielded a nearly 200-fold increase in IPTG-induced Cph1(Y176H)-sfGFP levels (20,100 ± 1,600 MEFL, Supplementary Fig. 3D). As expected, induction of *cph1(Y176H)** resulted in much stronger red fluorescence (848 ± 52 MEAP) in the presence of exogenously supplied PCB (Supplementary Fig. 3B).

We next integrated PPM v0.1 into our *cph1(Y176H)** strain, and measured red fluorescence to assay PCB production (Fig. 2, see "Methods" section). Unexpectedly, fluorescence remained below our limit of detection (Fig. 2). As with *cph1(Y176H)*, we suspected that *ho1* and/or *pcyA* may be poorly expressed leading to low PCB levels. Indeed, *sfgfp* translational fusions to these two enzymes resulted in low fluorescence values (246 ± 14 MEFL Ho1-sfGFP, 7.1 ± 1.6 MEFL PcyA-sfGFP) (Supplementary Figs. 4, 5) compared to a control strain where only *sfgfp* was expressed from $P_{rpsD}$ (1580 ± 190 MEFL) (Supplementary Fig. 6). Thus, we redesigned their expression cassette by separating both genes into independent transcriptional units, codon-optimizing the initial 15 codons of *ho1* (resulting in *ho1**) and the complete sequence of *pcyA* (resulting in *pcyA***), and introducing a transcriptional terminator downstream of *pcyA** to eliminate potential mRNA instability[45]. These changes resulted in a dramatic increase in expression of both enzymes (21,400 ± 1500 MEFL Ho1-sfGFP, 8350 ± 400 MEFL for PcyA-sfGFP) (Supplementary Figs. 4, 5). Thus, a new PPM v0.2 design was constructed based on these modifications (Fig. 2). However, PPM v0.2 produced very little red fluorescence (25.7 ± 6.2 MEAP) via the *cph1(Y176H)** PCB probe. This result suggested that high levels of Ho1 and PcyA were not sufficient for robust PCB production in *B. subtilis*.

Enzyme colocalization has been shown to increase yield in synthetic metabolic pathways, likely by generating high local concentrations of intermediate metabolites that overcome thermodynamically unfavorable steps[34,35]. To test whether colocalizing Ho1 and PcyA could increase PCB yields, we constructed PPM v0.3, wherein *ho1** is translationally fused to *pcyA*** via a flexible linker, and the fusion is again transcribed from $P_{rpsD}$ (see "Methods" section). Strikingly, fusing these enzymes resulted in an 80-fold increase in red fluorescence (2070 ± 130 MEAP, Fig. 2),

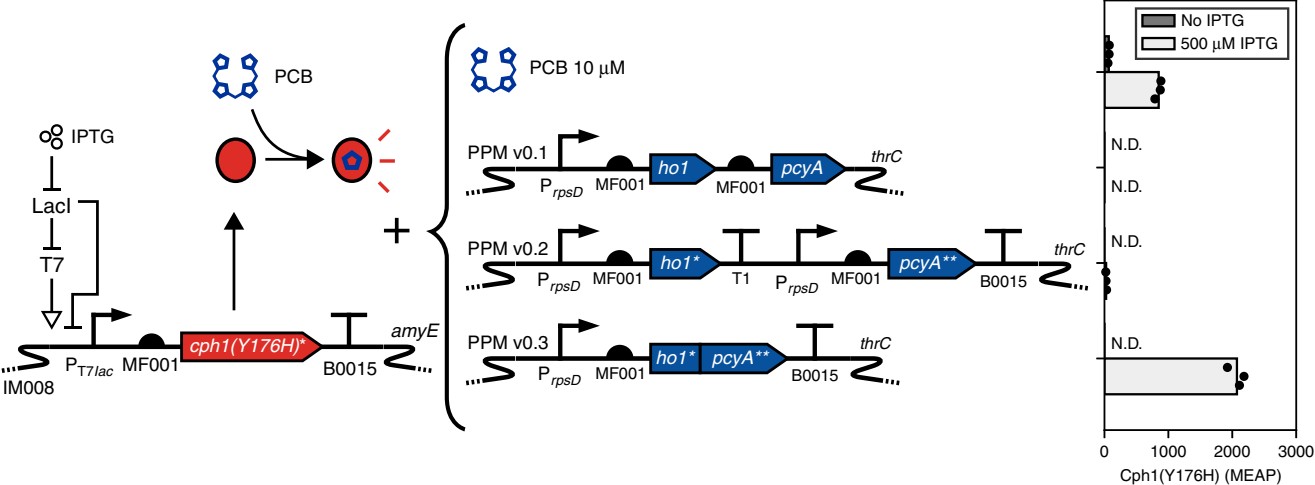

**Fig. 2** Debugging and optimizing the PCB production module. *B. subtilis* expressing *cph1(Y176H)** from LacI-T7[40] were grown in the absence or presence of IPTG, and in the presence of 10 μM exogenous PCB, or in combination with PPM v0.1, v0.2, or v0.3. The resulting red fluorescence of the Cph1(Y176H)-PCB conjugate was measured via flow cytometry. Bars show the mean of three experiments run on separate days. Black dots show values of individual experimental replicates. N.D.: not detected (see "Methods" section). Source data is available in the Source Data file

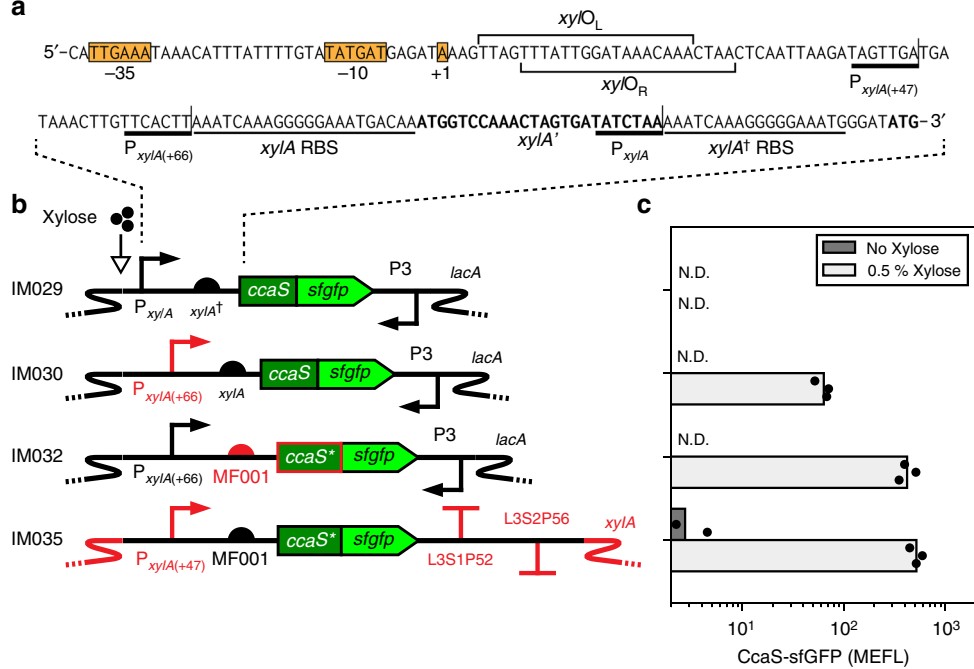

**Fig. 3** Debugging and optimization of the light-sensing module. **a** Annotated sequence of the legacy xylose-inducible promoter P$_{xylA}$ in LSM v0.1, obtained from integration plasmid pAX01[18,81]. The −35, −10, and +1 sites[82] and *xylR* operators xylO$_L$ and xylO$_R$[83] have been previously identified. The operators are followed by an unwanted untranslated region including the *xylA* RBS, and a vestigial truncated *xylA* ORF (*xylA'*). A second, truncated *xylA* RBS (*xylA†*) is present in pAX01 to enable translation of a gene of interest placed downstream. Brackets indicate the end of the legacy P$_{xylA}$ promoter, as well as sequential truncations P$_{xylA(+66)}$ and P$_{xylA(+47)}$, where problematic parts of P$_{xylA}$ are eliminated. The start codon of *ccaS* is shown at the end in bold. **b** Measurement and optimization of *ccaS* expression. We first inserted *ccaS-sfgfp* as in (**a**). Next, we truncated P$_{xylA}$ to remove the vestigial elements (resulting in P$_{xylA(+66)}$) and placed *ccaS-sfgfp* directly after the *xylA* RBS. Then, we switched the *xylA* RBS with synthetic RBS MF001 (see "Methods" section) and recoded the initial 15 codons of *ccaS*. Finally, we replaced the vestigial antisense promoter P3 with synthetic terminators L3S1P52 and L3S2P56[84], shortened the P$_{xylA(+66)}$ 5'UTR (resulting in P$_{xylA(+47)}$), and moved the entire cassette to the *xylA* chromosomal locus. **c** Expression of CcaS-sfGFP from each engineered module shown in (**b**). Bars show the mean of three experiments run on separate days. Dots show values of individual experimental replicates. N.D.: not detected (see "Methods" section). Source data is available in the Source Data file

which was more than double the fluorescence that occurred with exogenous PCB addition. Interestingly, levels of Ho1-PcyA-sfGFP from PPM v0.3 (19,900 ± 1100 MEFL, Supplementary Fig. 8) were comparable to those of the non-fused enzymes in PPM v0.2. Thus,

the dramatically enhanced PCB levels were not due to higher enzyme expression, but likely to more efficient transfer of BV from Ho1 to PcyA. We hypothesized that PPM v0.3 would support robust CcaSR function in *B. subtilis*.

**Debugging and optimizing CcaS expression**. We next utilized an *sfgfp* fusion to analyze *ccaS* expression from LSM v0.1. We observed no detectable fluorescence (Fig. 3), suggesting that *ccaS* is also very poorly expressed. A detailed analysis of the corresponding DNA sequence revealed several potentially disruptive elements downstream of $P_{xylA}$ that were inherited from our source material, the widely used plasmid pAX01[18]. In particular, the +1 site and *xylR* operators were followed by a 59-bp untranslated region (UTR) containing the *xylA* RBS, the first five codons of the *xylA* ORF, a stop codon that prematurely terminates this ORF, a second truncated copy of the *xylA* RBS, and an antisense promoter (Fig. 3). To eliminate any expression problems that could arise from these elements, we removed the vestigial ORF, the truncated *xylA* RBS, and the UTR after the +47 position, swapped in RBS MF001, recoded the initial *ccaS* ORF as before, and replaced the antisense promoter with terminators. Additionally, to eliminate the possibility of unwanted xylose consumption, we replaced the genomic xylose utilization operon with this new construct (LSM v0.2; Supplementary Fig. 9). As hoped, these changes resulted in a relatively large increase in CcaS expression (520 ± 74 MEFL) (Fig. 3, Supplementary Fig. 10). We hypothesized that LSM v0.2 would be sufficient for *B. subtilis* CcaSR function.

**Optimizing output promoter activity**. Strong maximal transcription is desirable for any engineered promoter system, including our optogenetic system. We next examined the activity of $P_{cpcG2-172}$ by comparing it to five *B. subtilis* constitutive promoters of different strengths under control of the house-keeping $\sigma^A$-RNA polymerase[36,46,47] (see "Methods" section). To measure promoter strength in a standardized manner, we designed a promoter characterization module wherein a self-cleaving ribozyme is followed by RBS MF001 and a *sfgfp* gene with an initial

codon-optimized sequence (*sfgfp\**). This design eliminates any differences in mRNA UTR sequences that may arise from the use of different promoters[48], thus ensuring that sfGFP fluorescence is a function of promoter strength alone (Supplementary Figs. 11, 12). While expression from the five reference promoters ranged from 1076 ± 100 MEFL to 133,000 ± 12,000 MEFL (Supplementary Fig. 11), maximal CcaR induction resulted in only 555 ± 65 MEFL sfGFP from $P_{cpcG2-172}$ (Fig. 4, Supplementary Fig. 12). Thus, $P_{cpcG2-172}$ is relatively weak in *B. subtilis*.

To increase $P_{cpcG2-172}$ activity, we first swapped the original −10 hexamer with the consensus *B. subtilis* sequence to enhance $\sigma^A$-RNA polymerase recruitment. Indeed, this modification increased output expression in the presence of fully induced CcaR nearly 3-fold (1470 ± 110 MEFL; Fig. 4). To further enhance activity, we then swapped the proximal region, or sequence following the −10 hexamer[49], with those from the two strongest promoters in our constitutive library, $P_{rpsD}$ and $P_{veg}$. Indeed, when *ccaR* is induced, both resulting chimeras ($P_{cpcG2-rpsD}$ and $P_{cpcG2-veg}$) exhibited very strong output (32,400 ± 2800 MEFL and 64,000 ± 7600 MEFL, respectively). As expected, we observed low reporter levels when *ccaR* is not induced (174 ± 23 MEFL and 378 ± 38 MEFL, respectively) (Fig. 4, Supplementary Fig. 13). Fold activation values of both promoters in response to IPTG (186.8 ± 9.3 and 169.1 ± 3.6) are not significantly different ($p = 0.10$, see "Methods" section). Because output gene expression from a strong promoter can always be brought down via many RBS tuning techniques[50,51], we chose to replace $P_{cpcG2-172}$ in TOM v0.1 with $P_{cpcG2-veg}$ (TOM v0.2) due to its superior activity.

**B. subtilis CcaSR v0.2 responds to light**. Next, we integrated PPM v0.3, LSM v0.2, and TOM v0.2 into the chromosome of a single *B. subtilis* strain, resulting in *B. subtilis* CcaSR v0.2. As before, we measured green fluorescence from cultures grown at

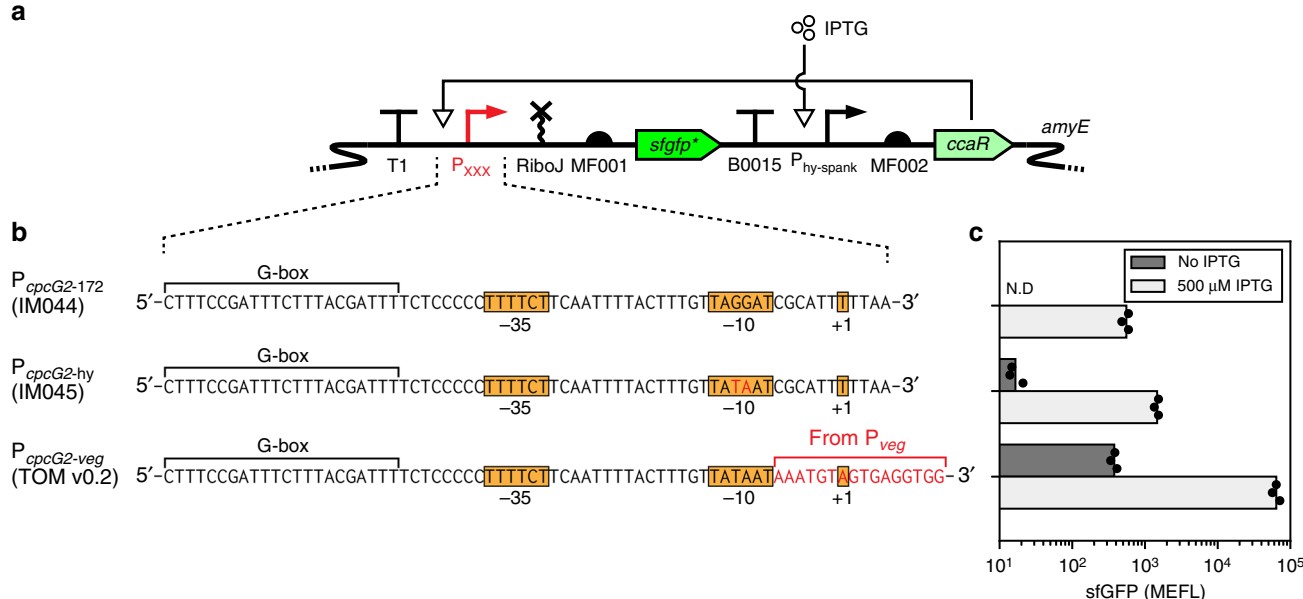

**Fig. 4** Optimization of the transcriptional output module. **a** Characterization of $P_{cpcG2-172}$ and subsequently-engineered variants using a standard promoter characterization cassette. The promoters are inserted downstream of a transcriptional terminator to prevent any read-through transcription from the *amyE* locus, and drive transcription of an mRNA containing the self-cleaving ribozyme RiboJ[48], MF001, and *sfgfp* with the first 15 codons optimized (*sfgfp\**) (see "Methods" section). IPTG induces *ccaR* expression, and low-levels of CcaR phosphorylation, likely from a small-molecule phosphoryl donor such as acetyl phosphate[37], should activate transcription from $P_{cpcG2-172}$ and further variants. **b** Regions of interest of $P_{cpcG2-172}$, $P_{cpcG2-hy}$ (named after $P_{spac-hy}$, a mutated stronger version of IPTG-inducible *B. subtilis* promoter $P_{spac}$[85]), and $P_{cpcG2-veg}$. Modifications performed at each step are highlighted in red. The putative binding region for CcaR (G-box) is indicated[26]. **c** sfGFP fluorescence resulting from each output promoter system shown in (**b**) in the absence and presence of saturating IPTG. Bars show the mean of three experiments run on separate days. Dots show values of individual experimental replicates. N.D.: not detected (see "Methods" section). Source data is available in the Source Data file

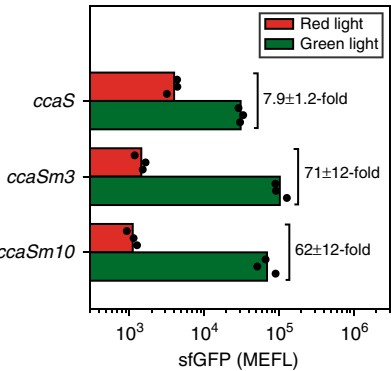

**Fig. 5** CcaS miniaturization increases light response. *B. subtilis* expressing PPM v0.3, TOM v0.2, and either LSM v0.2 (*ccaS*), LSM v0.3a (*ccaSm3*), or LSM v0.3b (*ccaSm10*) were grown at 48 combined levels of IPTG and xylose, under saturating red or green light. sfGFP fluorescence values shown here correspond to the inducer concentrations that resulted in the highest fold-change in response to light for each *ccaS* variant (Supplementary Figs. 16, 19, 20). Bars show the mean of three experiments run on separate days. Dots show values of individual experimental replicates. Source data is available in the Source Data file

48 combined xylose and IPTG levels under both red and green light (Supplementary Figs. 14, 16). First, we observed that sfGFP fluorescence increased with *ccaR* induction, as expected. Additionally, in both red and green light, *ccaS* induction resulted in an initial increase followed by a large decrease in sfGFP output. We previously observed similar profiles in *E. coli*, likely due to limiting PCB at high CcaS levels, resulting in Apo-CcaS proteins with phosphatase activity[28]. For a wide range of *ccaS* and *ccaR* induction levels, sfGFP levels were higher under green than red light, with a maximum fold-change of $7.9 \pm 1.2$ at 0.0106% xylose (68.7 MEFL CcaS-sfGFP) and 10.0 μM IPTG (537 MEFL CcaR-sfGFP) (Supplementary Figs. 14, 16). A control strain lacking PPM v0.3 did not respond to light (Supplementary Figs. 15, 17). Based on these results, we conclude that *B. subtilis* CcaSR v0.2 functions properly, albeit with low dynamic range.

**CcaS miniaturization increases dynamic range**. To increase the dynamic range of *B. subtilis* CcaSR v0.2, we separately replaced *ccaS* in LSM v0.2 with *mini-ccaS#3* and *mini-ccaS#10* (hereafter *ccaSm3* and *ccaSm10*), two miniaturized variants that exhibit enhanced dynamic range in *E. coli*[29,30]. Indeed, both of the variants increased dynamic range approximately 10-fold (*ccaSm3*: $71 \pm 12$-fold, *ccaSm10*: $62 \pm 12$-fold) (Fig. 5, Supplementary Figs. 19, 20). In contrast to *E. coli*, where *ccaSm10* resulted in a much larger dynamic range, both variants give rise to statistically identical values in *B. subtilis* ($p = 0.49$, see "Methods" section). In both organisms, however, *ccaSm10* results in overall lower output expression levels compared to *ccaSm3*. The reason why this only results in a larger fold-change in *E. coli* is currently unknown. Due to its superior performance, we named the system containing *ccaSm3 B. subtilis* CcaSR v1.0, and carried it forward for further characterization.

**Characterization of *B. subtilis* CcaSR v1.0 Light Response**. To compare *B. subtilis* CcaSR v1.0 activity to that of our *E. coli* systems, we measured the steady-state transfer function (Fig. 6b) by growing the engineered strain under varying intensities of green light (see "Methods" section). With increasing green light intensity, sfGFP gradually increased from $2000 \pm 190$ to $108,700 \pm 9200$ MEFL in a manner well approximated by a Hill function. Fits to the experimental data revealed a Hill coefficient

of $1.88 \pm 0.16$ and a 50% activation intensity of $4.66 \pm 0.63$ μmol m$^{-2}$ s$^{-1}$ (Fig. 6b). Thus, *B. subtilis* CcaSR v1.0 exhibits analog green light-intensity dependent transcriptional output, as our *E. coli* systems do. Interestingly, this transfer function was more gradual and the 50% activation intensity was more than four times higher than in the *E. coli* system (Hill coefficient: $2.737 \pm 0.044$, 50% activation intensity: $1.075 \pm 0.025$ μmol m$^{-2}$ s$^{-1}$)[30]. The origins of these differences are not immediately clear and warrant further investigation.

Finally, we characterized the response dynamics of CcaSR after an instantaneous switch from saturating red light to green light (Fig. 6c) or vice-versa (Fig. 6d). Both responses showed a short delay followed by an exponential-like increase or decrease in sfGFP fluorescence until saturation, with half-maximum activation ($t_{1/2}$on) and deactivation ($t_{1/2}$off) times of $105.1 \pm 1.5$ and $74.97 \pm 0.39$ min, respectively (see "Methods" section). These response times are similar to those observed in *E. coli*, albeit slightly slower ($t_{1/2}$on = 77 min, $t_{1/2}$off = 62 min)[30]. To explain this difference, we note that in our previous *E. coli* work we have used an antibiotic protocol to arrest cell growth and protein synthesis and allow maturation of all intracellular sfGFP[20]. Because all sfGFP was mature at the time of measurement, the effect of sfGFP maturation kinetics in the observed dynamics was eliminated. In contrast, we have been unable to develop a similar *B. subtilis* maturation protocol. Therefore, sfGFP maturation kinetics slow the observed response dynamics[52], especially as expression increases and more immature sfGFP is produced. Thus, these results show that CcaSR exhibits similar dynamic I/O properties in *B. subtilis* and *E. coli*.

## Discussion

We have engineered CcaSR as the first optogenetic tool for *B. subtilis*, the leading model Gram-positive bacterium. CBCRs, and their bilin-binding relatives the phytochromes, tend to respond to lower intensities of light than other commonly used optogenetic tools, such as the flavin-binding LOV domain or Cryptochrome 2[53]. Although slightly greater light intensities are required to activate CcaSR in *B. subtilis* as compared to *E. coli*, they are still within the range of intensities used to modulate most other bacterial optogenetic tools[54–57]. Furthermore, these do not lead to phototoxic effects in *B. subtilis* (Supplementary Fig. 21).

One of the challenges when porting a bilin-binding photoreceptor into a heterologous host organism is chromophore availability. PcyA has evolved to utilize cyanobacterial or plant ferredoxins to produce PCB, and it has been shown to work poorly with Gram-positive ferredoxins[58]. Recently, two groups achieved efficient PCB production in mammalian cells by expressing a plant ferredoxin and a ferredoxin-NADPH reductase in addition to *ho1* and *pcyA*[59,60]. In contrast, here we showed that a translational *ho1-pcyA* fusion can dramatically increase PCB production without additional enzymes. Theoretical models suggest that colocalization alone should not improve yield unless multiple copies of each enzyme are clustered[35,61], or the enzymes naturally engage in substrate channeling, the transfer of intermediate metabolites via protein-protein interaction[62]. Interestingly, channeling may naturally occur between human heme oxygenase and biliverdin reductase in the human heme catabolic pathway[63]. Our results are consistent with Ho1 and PcyA engaging in substrate channeling, where the flexible linker increases the rate of BV transfer and therefore the metabolic flux. Translational fusions may be a general method to increase biosynthetic pathway yields in cellular environments where some accessory proteins are lacking.

As in *E. coli*, CcaSR should enable unrivaled temporal and spatial control of gene expression in *B. subtilis*. In particular, by

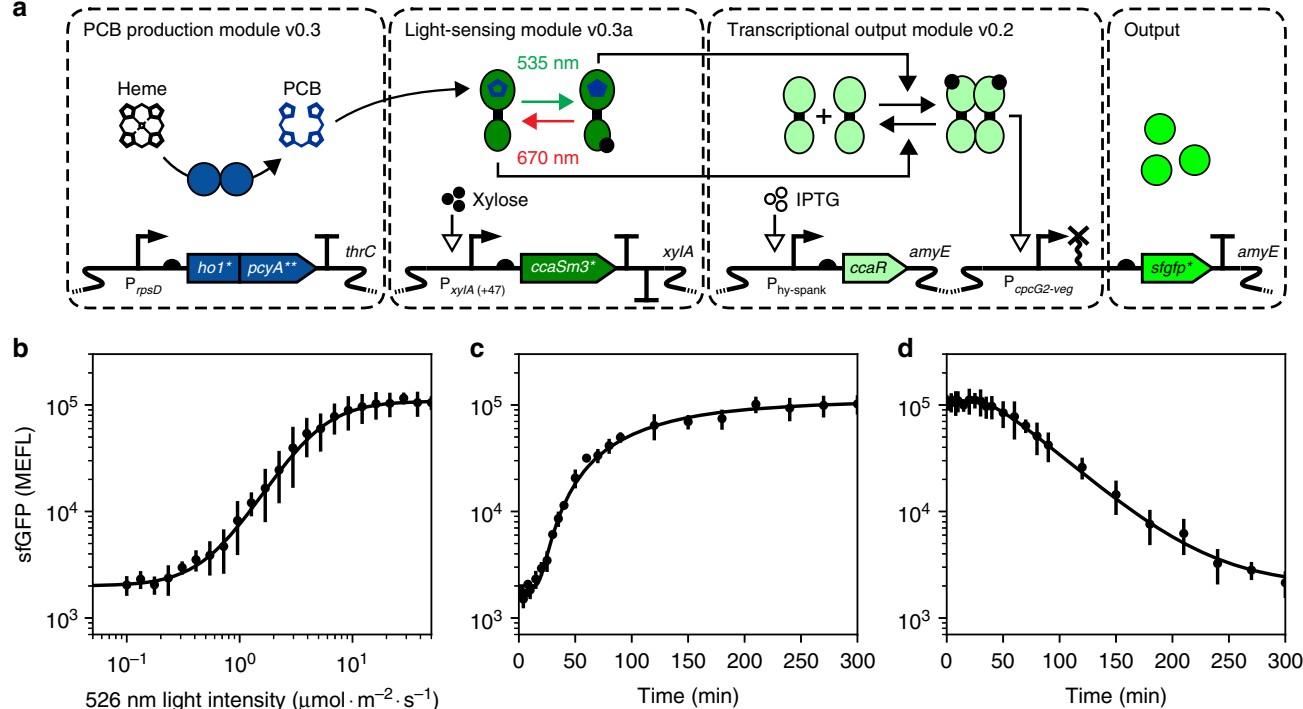

**Fig. 6** Characterization of *B. subtilis* CcaSR v1.0 input/output properties. **a** Schematic of the final *B. subtilis* CcaSR v1.0 design. **b** Steady-state transfer function. **c** Step ON response. **d** Step OFF response. Dots and error bars show the mean and standard deviation, respectively, of three experiments run on separate days. Black lines represent model fits (see "Methods" section). Source data is available in the Source Data file

conducting additional step ON and step OFF kinetic measurements at different initial and final light intensities, we should be able to construct a general mathematical model of how CcaSR output gene expression changes over time in response to any time-varying light input signal, following our established methods[20]. Then, we can utilize our biological function generator method to design light signals capable of driving sophisticated gene expression signals with quantitative accuracy[20].

By encoding regulatory proteins as the output of CcaSR, such signals can be used as a tool to interrogate the dynamical signal processing properties of important cellular decision-making networks such as those involved in stress response, sporulation, and biofilm formation. For example, while the mechanism by which pulses of the master sporulation regulator Spo0A arise has been elucidated[8,9], their impact on downstream processes remains unclear. Interestingly, the use of a chemically-inducible promoter to express the constitutively-active mutant Spo0A-sad67 does not trigger sporulation[64]. It has been proposed that fast Spo0A activation leads to early repression of genes that are essential for sporulation, thus leading to non-viable spore formation[65]. CcaSR could be used to create different dynamical expression patterns of Spo0A-sad67, such as ramps of different slopes and pulses of different amplitudes, frequencies, or phases, in order to understand how the downstream sporulation circuit responds to Spo0A activity dynamics. Moreover, the ability to use light to control gene expression in individual cells could be exploited when studying processes involving noisy or spatially-delineated gene expression patterns.

We have recently characterized the full spectral response properties of CcaSR and the related red/far-red photoreversible sensor Cph8-OmpR to overcome inherent spectral overlap between the systems and achieve simultaneous and independent control of the expression dynamics of two separate genes in *E. coli*[32]. We believe Cph8-OmpR or other bacterial optogenetic

tools[54,55,57,66] could also be ported into *B. subtilis* and combined with CcaSR to achieved multiplexed control of gene expression dynamics. Such a technology would be particularly useful in examining whether *B. subtilis* gene circuits require specific combinations of dynamical gene expression signals to function properly[22].

This work also advances *B. subtilis* synthetic biology in several ways. First, our results suggest that genes ported into *B. subtilis* from other organisms may frequently be inefficiently translated due to mRNA secondary structure between the RBS and initial ORF sequence. Since strong *B. subtilis* RBSs seem to have little sequence variation[43], RBS redesign to reduce secondary structure might not always be feasible (Supplementary Fig. 22). On the other hand, several studies have shown that strongly expressed bacterial genes have an initial codon bias that minimizes secondary structure[41–43]. Furthermore, secondary structure entirely contained within the initial ORF sequence has been found to greatly affect translation as well[44]. All cyanobacterial genes in this study, apart from *ccaR*, suffered initially from poor expression. In all cases, we showed that codon optimization of the initial 15 amino acids can result in dramatic expression improvements (Supplementary Fig. 23). A computational tool that modifies an ORF sequence to tune translation currently exists only for *E. coli*[67]. Our results indicate that a tool for *B. subtilis* that operates under the same principles is feasible. However, validation against a large variety of genes and expression cassettes would be necessary.

We also show that proper insulation from the genomic context, in the form of a transcriptional terminator at the 3' of an expression cassette, leads to a dramatic increase in gene expression in at least two of the most widely-used genomic integration loci: the *thrC* locus (Supplementary Fig. 5, 21-fold PcyA expression difference) and the *amyE* locus (Supplementary Fig. 7, 11-fold sfGFP expression difference). While at least one study

recognizes the need for a well-insulated expression cassette in *B. subtilis*[46], plasmids for C-terminal fusions to *lacZ* or other reporter genes offered by the Bacillus Genetic Stock Center are not properly insulated. It is possible that incorporation of a single terminator could greatly improve the limit of detection of future assays based on these plasmids.

The sequence determinants of the strength of promoters controlled by the housekeeping sigma factor $\sigma^A$ are classically thought to be the −35 and −10 hexamers, and to a lesser extent a −16 region and a A/T rich sequence upstream of the −35 hexamer[68–70]. Here we show that the sequence downstream of the −10 hexamer can greatly affect transcription as well. We found that $P_{cpcG2-172}$ output was low compared to a small library of constitutive promoters. To engineer a stronger promoter while keeping the CcaR operators intact, we swapped the sequence after the -10 hexamer with sequences from strong constitutive promoters $P_{rpsD}$ and $P_{veg}$. These mutations increased output transcription by 22- and 44-fold, respectively. A larger scale study to explore the generality of this technique would be of great interest to *B. subtilis* synthetic biology practitioners, particularly of those that work in engineering transcription factor-inducible promoters, where the transcription factor binding site covers everything but the sequence after the −10 hexamer.

In conclusion, we have engineered the first, to our knowledge, *B. subtilis* optogenetic system. *B. subtilis* CcaSR v1.0 should enable precise temporal and spatial control of gene expression in *B. subtilis*, and thus be widely used to study complex cellular processes such as sporulation, stress response, and biofilm formation, among others. We have also unveiled several design principles that should be of great utility in future *B. subtilis* synthetic biology applications.

## Methods

**DNA and strain construction**. All cloning and experiments were performed in *B. subtilis* strain PY79. A list of all strains constructed in this study can be found in Supplementary Data 1. Strains with each one of the final CcaSR modules (PPM v0.3, LSM v0.3a, TOM v0.2), the complete CcaSR v1.0 system, and the optimized *cph1(Y176H)\** PCB-dependent fluorescent protein can be obtained from the Bacillus Genetic Stock Center via the specified accession numbers. All other strains are available from the corresponding author on reasonable request.

Primers were ordered from Integrated DNA Technologies, Inc. All TCS genes were amplified from previous *E. coli* CcaSR plasmids[28]. Cph1(Y176H) was amplified from the Cph1-EnvZ chimera Cph8 in pSR33.4[28]. *pcyA\*\** was designed using GeneOptimizer[71] and ordered as a gBlock from Integrated DNA Technologies, Inc. The xylose and IPTG-inducible cassettes were amplified from integration plasmids pAX01[18] and pDR111[19]. Constitutive promoters were constructed via oligo annealing and extension. Sequences of $P_{liaG}$, $P_{lepA}$, and $P_{veg}$ comprise bases from the −66 to the +10 ($P_{liaG}$, $P_{veg}$) or +12 ($P_{lepA}$) position of the corresponding promoters of the BioBricks library[47]. The sequence of $P_{rpsD}$ comprises bases from the −58 to the +16 position of the corresponding chromosomal promoter[36]. $P_{S1*}$ is identical to $P_{sarA}$-derived $P_{S1}$[46], but with the unmodified $P_{sarA}$ 5'UTR. Synthetic RBS MF001 was obtained from integration plasmid pMF35[72]. Genomic homology fragments required for chromosomal integration were amplified from the purified genome of *B. subtilis* PY79. A list of genetic parts, along with their sequences, can be found in Supplementary Data 2.

All systems were built as linear double-stranded integration modules (IMs)[73]. The IMs contain the DNA of interest and a selection marker flanked by 1.5 kb-long sequences homologous to a region of the *B. subtilis* genome (integration locus) where chromosomal integration via double crossover occurs. IMs were assembled from PCR-amplified parts using GoldenGate[74]. The resulting Golden Gate products were amplified using NEB Phusion DNA Polymerase and gel purified to obtain the IM. 500 ng or more of each IM was transformed into competent *B. subtilis* using standard transformation methods. The transformants were plated on selective media. Colonies picked the next day and grown in LB media at 37 °C and 250 RPM for ~2 h until turbidity is visible. Finally, freezer stocks were prepared with 700 µL culture and 300 µL 60% glycerol, and stored at –80 °C. This method avoids subcloning of integration plasmids in *E. coli*, as long as enough PCR-amplified DNA can be obtained. A list of all IMs constructed in this study can be found in Supplementary Data 3, and their complete sequences can be found on genbank via the specified accession numbers.

For sequence verification, an overnight LB culture was grown from a freezer stock, and 2 µL saturated culture was used as template for a 50 µL PCR reaction,

either with Taq or Phusion DNA Polymerase. PCR products obtained in this fashion were gel-purified and sent for sequence verification to Genewiz, Inc.

To construct strains with two IMs, two separate strains containing each IM were cloned and sequence-verified independently, as described above. Next, one of these strains was made competent, and the genomic DNA of the second was extracted from an overnight culture using the Promega Wizard Genomic DNA Purification Kit. Next, 500–1000 ng of purified genomic DNA was transformed into the competent strain as described above to produce a new strain with both IMs. Correct integration of both modules was verified via PCR. If necessary, a strain with two IMs was made competent and a third IM was integrated via the same procedure.

**Codon optimization**. For each of the first 15 codons, a synonymous codon was chosen to reduce GC and increase AU content, with A preferred over U, with no regard for codon frequency. The free energy of the resulting mRNA secondary structure was calculated via Nupack[75] using the sequence from the transcription start site up to the 90th nucleotide residue of the ORF.

**Media and experimental protocols**. We used a modified M9 media for experiments. 1 L 5× M9 salts at pH ~6.8 were prepared with 64 g $Na_2HPO_4.7H_2O$, 15 g $KH_2PO_4$, 2.5 g NaCl, 5 g $NH_4Cl$, 9.2 mL 6 M HCl, and up to 1 L $dH_2O$. For 1 L M9, we used 200 mL 5× M9 salts, 20 mL 10% casamino acids, 6.67 mL 60% glycerol, 1 mL 50 mM $FeCl_3$/100 mM $C_6H_8O_7$ solution, 2 mL 50 mM $MnSO_4$, 2 mL 1 M $MgSO_4$, 100 µL 1 M $CaCl_2$, and $dH_2O$ up to 1 L. Glycerol was used as a carbon source since glucose strongly represses expression from $P_{xylA}$. The $FeCl_3$ solution appears necessary to support robust growth of PCB-producing strains.

For every experiment, an overnight LB culture was started from the freezer stock of each relevant strain. The next day, saturated cultures (OD600 ~3) were diluted $10^4$–$10^5$-fold in M9. For non-optogenetic experiments, media was distributed in culture tubes (3 mL per tube), inoculated with the appropriate inducers, and incubated in a shaker operating at 250 RPM and 37 °C until the $OD_{600}$ reached between 0.08 and 0.15 (6–9 h depending on initial density and strain). When indicated, saturating concentrations of xylose (0.5%) and IPTG (500 µM) were used. Inducer concentrations used in Fig. 6 were 0.00917% xylose and 10.0 µM IPTG. In optogenetic experiments, media was distributed (500 µL per well) in 24-well dark-walled clear-bottomed plates (ArcticWhite AWLS303008). Next, the appropriate inducers were added, and plates were sealed with adhesive foil (VWR 60941-126). Plates were mounted onto LPAs running the appropriate light exposure programs, and incubated until the $OD_{600}$ reached between 0.08 and 0.15. Culture tubes or plates were then transferred to ice. 100 µL of each sample was transferred to a flow cytometry tube containing 1 mL PBS for measurement.

PCB was purified from Spirulina powder using methanolysis, as previously described[76]. Samples, where purified PCB was added, were incubated in culture tubes at 250 RPM and 37 °C for 5 h. Next, 10 µM purified PCB was added under a green safelight, and tubes were returned to the incubator for 1 h. At the end of the experiment, tubes were placed on ice and Cph1(Y176H) fluorescence was measured via flow cytometry.

**Optical hardware**. Eight 24-well Light-Plate Apparatuses (LPAs)[77] equipped with green (520-2-KB, WP7083ZGD/G, Kingbright, CA, USA) and red (660-LS, L2-0-R5TH50-1, LEDSupply, VT, USA) LEDs were used for all optogenetic experiments. These were mounted in a shaking incubator operating at 250 RPM and 37 °C. Total LED power output in µmol s$^{-1}$ was measured using a spectrometer (StellarNet UVN-SR-25 LT16) attached to a six-inch integrating sphere (StellarNet IS6). The average light intensity was calculated by dividing the total power output by the area of a circular plate well with a radius of 7.5 mm, as previously[32]. To calibrate the power output of each LED, we adjusted the current using the LPA Dot Correction setting to achieve saturating intensities of 20 µmol m$^{-2}$ s$^{-1}$ or more for red LEDs and 50 µmol m$^{-2}$ s$^{-1}$ or more for green LEDs. Each LED was measured while powered from the same LPA socket used in experiments. The precisely measured intensities and dot correction values were recorded. Custom Python scripts were written to use these recordings to achieve light intensities necessary in each experiment.

**Flow cytometry analysis**. The sfGFP fluorescence distribution of each *B. subtilis* culture sample was measured using a BD FACScan flow cytometer with an excitation source of 488 nm and an emission window of 510/21 nm. 10,000-30,000 events were collected per sample. A suspension of calibration beads (Spherotech RCP-30-5A) in PBS was measured with each experiment. After data acquisition, raw.fcs flow cytometry files were processed using FlowCal[78]. Cell populations were gated by forward scatter/side scatter density (Supplementary Fig. 24), retaining 50% of the total number of events. Next, fluorescence of each gated event in arbitrary units was converted into standardized MEFL (Molecules of Equivalent Fluorescein) values using the calibration bead data. The cell fluorescence of each culture sample was then obtained from the median MEFL fluorescence of all gated events in that sample. Finally, the sfGFP fluorescence was obtained by subtracting the cell fluorescence of each sample from the cell fluorescence of a control sample lacking any fluorescent reporter gene measured the same day (autofluorescence control). The strain used as an autofluorescence control for any strain containing

PPM v0.3 was a strain containing the PPM v0.3 only. The autofluorescence strain for any other strain was the wild-type PY79.

The single cell Cph1(Y176H) fluorescence distribution was measured using a BD FACSCanto II. The fluorescent channel used (APC-A) used a 633 nm red laser as the excitation source, and an emission window of 670/50 nm. Calibration beads were measured as above. FlowCal was used to gate cell populations and calibrate fluorescence data to MEAP (Molecules of Equivalent Allophycocyanin). The cell fluorescence of each culture sample was then obtained from the median MEAP fluorescence of all gated events in that sample. Finally, the Cph1(Y176H) fluorescence was obtained by subtracting the cell fluorescence of each sample from the cell fluorescence of a control wild-type PY79 sample.

**Statistical analysis.** Each experiment was replicated three times over different days. Fluorescence of each sample is reported as the mean ± std dev. of the sfGFP and Cph1(Y176H) fluorescence from three experiments. A one-sample Student's $t$-test was conducted for every sample to evaluate whether sfGFP or Cph1(Y176H) fluorescence was significantly different from zero (one-sided, $p < 0.05$). Fluorescence of samples that failed this test are reported as not detected or N.D. Significance in the difference of fold changes was calculated via a two-sample Welch's unequal variances $t$-test.

**Transfer function modeling and fitting.** Steady-state transfer functions in Fig. 6b, Supplementary Figs. 14, 15, and 18 were fitted to a Hill Function of the form:

$$y = y_0 + \Delta y \cdot \frac{x^n}{x^n + K_{1/2}^n}$$

Here, $y$ is the observed sfGFP fluorescence in MEFL, which has a minimum value of $y_0$ in the absence of inducer and a maximum of $y_0 + \Delta y$ under saturating conditions, $x$ is either light intensity in $\mu$mol m$^{-2}$ s$^{-1}$ (Fig. 6b), Xylose concentration in % (Supplementary Figs. 14B, 15B, and 18) or IPTG concentration in $\mu$M (Supplementary Figs. 14D and 15D), $K_{1/2}$ is the inducer concentration for half-maximum activation, and $n$ is the Hill coefficient. Fitting was performed using the LmFit python package[79] with the Levenberg-Marquardt algorithm. Experimental data from three replicates were combined and fitted simultaneously. To adequately fit low and high fluorescence values, the error to minimize was defined as the difference between the logarithm of a fluorescence datapoint and the logarithm of the model prediction. Fitted parameter values and their uncertainties can be found in Supplementary Data 4.

**Kinetic response modeling and fitting.** The response of the CcaSR system to an instantaneous change in light intensity was modeled as a differential equation system of the form:

$$\frac{dp}{dt} = k_p \cdot \left( c\left(I_g, I_r\right) - p(t) \right)$$

$$\frac{dg}{dt} = k_g \cdot \left( p(t - \tau) - g(t) \right)$$

$$\frac{dG}{dt} = k_d \cdot \left( g(t) - G(t) \right)$$

Here, $p(t)$ represents the sfGFP production rate, $g(t)$ represents immature sfGFP, and $G(t)$ is the fully mature, observed sfGFP. Their dynamics are determined by rate constants $k_p$, $k_g$, and $k_G$. $\tau$ represents the delay before a change in sfGFP production actually occurs. Finally, $c$ is the system input, and is a function of light intensities $I_g$ and $I_r$. Units for $c$, $p(t)$, and $G(t)$ have been chosen such that, in steady state, $c = p_{ss} = g_{ss} = G_{ss}$, and thus $c$ determines the steady state output fluorescence. To simulate an instantaneous change from red to green light (Fig. 6c), the model was solved with initial conditions $p(0) = g(0) = G(0) = y_0$ and $c = y_0 + \Delta y$, where $y_0$ and $\Delta y$ are obtained from the Hill function fit. Similarly, an instantaneous change from green to red light (Fig. 6d) was simulated with $p(0) = g(0) = G(0) = y_0 + \Delta y$ and $c = y_0$. Fitting was performed using LmFit as described above.

**Reporting summary.** Further information on research design is available in the Nature Research Reporting Summary linked to this article.

## Data availability
Sequences of all integration modules used in this study are available from Genbank via the accession numbers listed in Supplementary Data 3. Flow cytometry data and scripts used to generate all figures are available from figshare (https://doi.org/10.6084/m9.figshare.8198999)[80]. Parameters of Hill function fits can be found in Supplementary Data 4. Source data for Figs. 2–6 and Supplementary Figs. 1–8, 10–22 can be found in the Source Data file. Any other relevant data can be obtained from the authors upon reasonable request.

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

## Acknowledgements

We thank Karl Gerhardt for performing the PCB purification protocol. This work was supported by a Michel Systems Biology Innovation Award from Rice University, the National Institutes of Health (1R21AI115014-01A1), and the National Science Foundation (DMR 1611044). O.A.I. also acknowledges support from the National Science Foundation (MCB-1616755) and the Welch Foundation (Grant C-1995).

## Author contributions

S.M.C., M.F., O.A.I., and J.J.T. conceived of the project. S.M.C. designed and built DNA constructs. S.M.C. and E.A.B. collected data. S.M.C. performed all data analyses. O.A.I. and J.J.T. supervised the project. S.M.C., M.F., O.A.I., and J.J.T. wrote the manuscript.

## Additional information

**Competing interests:** The authors declare no competing interests.

