## [Peer Review File · Nature Communications]

Reviewers' Comments:

Reviewer #1:

Remarks to the Author:

In the present work, Tabor and coworkers port their trademark CcaSR system for red/green light dependent gene expression to *Bacillus subtilis*. In doing so, they enable previously unattainable optogenetic regulation of cellular events in this widely used, gram positive model bacterium. Initial attempts at adapting previous versions of the CcaSR system, earlier established in *E. coli*, were fraught with problems that could eventually be traced to poor promoter performance in the heterologous host and inefficient production of the phycocyanobilin (PCB) chromophore in *B. subtilis*. These problems were specifically addressed and thus overcome to achieve the desired light sensitivity of expression. The replacement of the original CcaS with a recently reported miniaturized version enhanced the regulatory degree of the system. The manuscript winds up with a quantitative analysis of the light response in the best performing system.

Overall, this manuscript reports an exercise in synthetic biology that eventually yields a new light responsive circuit which should be of utility to researchers working on *B. subtilis*. Although the work clearly is an extension of Tabor's ample previous work on the same optogenetic circuit, it possesses certain novelty and merit in that to date no reliable setup for light dependent expression control in *Bacillus* has been reported. That said, much of the manuscript amounts to a rather lengthy and often overly detailed narrative of how the optimized system was constructed. It is unclear how much general interest exists in these aspects which could easily have been subsumed and/or relegated to the supplementary material. It is to be lamented that the authors settle for characterizing the response dynamics of their best system using a reporter gene but do not go beyond and apply it for resolving a biological question: clearly the focus of this work is on implementing a new method. A possible application to *Bacillus* sporulation dynamics is mentioned but not followed up on. The authors purport that their findings generally apply to the genetic manipulation of *B. subtilis* but this claim appears somewhat unsubstantiated given that a single genetic system was studied. Parts of the manuscript are hard to grasp on first reading because of the technical lingo and many abbreviations used, some of which are only introduced much later in the manuscript (eg MEFL, MEAP, APC). Other than that, the manuscript is well written, and the methods are described in detail and completeness. Taken together, this manuscript is of interest to a specialized readership but its general appeal remains in doubt.

Reviewer #2:

Remarks to the Author:

The manuscript by Castillo-Hair, Baerman, Fujita, Igoshin, and Tabor adapts the CcaS-CcaR green/red optogenetic system, which they previously optimized in *E. coli*, for use in the Gram positive bacterium *B. subtilis*. The authors systematically debug the optogenetic system to achieve a dynamic range of over 70-fold. The authors interrogate each 'module' of the two component system (chromophore production, light sensing, output actuation) for failure points. Through this process, they both dramatically improve the function of CcaSR in *B. subtilis* and reveal principles that have the potential to be generalizable to other genetic engineering projects in *B. subtilis*. The authors increase transcriptional and translational efficiencies in addition to enhancing chromophore (PCB) production through a translational fusion of synthesis enzymes, an improvement that may translate to other bilin-binding systems. Once debugged, the authors establish light intensity and ON/OFF temporal responses of this system. Optogenetic study of this model Gram positive bacterium will allow for temporal and spatial control of regulatory processes to build upon and enhance static control studies. The experiments described in the manuscript are of high technical quality and the article itself is clearly written. Given the potential impact that introduction of precise, optogenetic control contributes to the toolbox for *B. subtilis* researchers, I am enthusiastic about these results. It will be very interesting to see this tool applied to tackle problems such as the sporulation application described in the Discussion, among others.

Minor Comments:

1. Phototoxicity effects are not currently quantified in the manuscript. Comparing the light intensity curve (Fig. 6B) with that of the version 3 CcaSR in *E. coli*, the maximum steady state output is reached with an order of magnitude greater light intensity. It would be helpful to include data supporting the claim in line 294 that phototoxic effects are not a concern.
2. Regarding the PcpG2-rspD vs PcpG2-veg (Fig. 4 vs Fig. S12): Why is the veg variant superior? It seems that the rspD has a higher fold change and lower leakage. What is the rationale for selecting veg?
3. CcaSm3 performs better in this study, but ccaSm10 is superior in the version 3 paper in *E. coli*. Please comment on any potential reason for this difference. (Fig. 5).
4. In Fig. S5 the authors claim that the added terminator accounts for most of the increase in expression. However, they lack a construct that has no terminator and codon optimized pcyA^{**}. This point is relevant in the context of providing information for future *B. subtilis* synthetic biology efforts. It would be useful to know what the impact is on full codon optimization in isolation of the terminator effects.
5. Fig. S20: sfgfp* (15 bp codon optimized) does show higher expression (B). However, the unpaired probability seems to be lower than that of the regular sfgfp (C vs D). This seems potentially inconsistent with the explanation about why codon optimization increases expression. Please clarify.
6. Line 501: These are three first order differential equations, not third order differential equations.
7. Fig. 2 caption typo: "absence of presence"
8. Fig. 4: In this setup, how would CcaR be phosphorylated to act on the PcpG2 variants (as these are in the absence of CcaS, if I understand correctly)?
9. Line 112: This is a very minor suggestion to improve readability, but it took me a minute to figure out what the "M" was in PPM, etc. so I might suggest moving this to read as "PCB production module", etc.
10. Line 222: sfgfp* is introduced in the text, but it isn't clear what this is. I did eventually find it in the figure caption.
11. Line 395: please update "for a few hours" to use more precise language.
12. Methods: inconsistent capitalization on rpm vs. RPM.

Author responses to reviewer comments

Reviewer #1

Much of the manuscript amounts to a rather lengthy and often overly detailed narrative of how the optimized system was constructed. It is unclear how much general interest exists in these aspects which could easily have been subsumed and/or relegated to the supplementary material.

Thank you for this suggestion. Looking back at the manuscript, we realized that the section on optimization of the PCB production module was much larger than later sections on optimization of other modules (7 paragraphs compared to 1 for the light-sensing module and 2 for the transcriptional output module) and described every optimization step in a level of detail that was not consistent with later sections. This was motivated by the fact that many subtasks needed to be described (*cph1*(Y176H) validation, enzyme expression optimization, chimeric *ho1-pcyA* construction and validation). However, it is true that some of this detail could have been simplified and/or moved into the SI.

In this revised version, we have reduced this section to 4 paragraphs. Specifically, we simplified the description of expression optimization of *ho1* and *pcyA*. Furthermore, we moved Figures 2B and 2C, originally describing optimization of enzyme expression, into Supplementary Figures 4 and 5, where individual optimization steps are described in more detail. We think these modifications should improve the manuscript.

It is to be lamented that the authors settle for characterizing the response dynamics of their best system using a reporter gene but do not go beyond and apply it for resolving a biological question: clearly the focus of this work is on implementing a new method. A possible application to *Bacillus* sporulation dynamics is mentioned but not followed up on.

We are indeed using CcaSR to study *B. subtilis* sporulation dynamics. However, that is substantial additional work, and we believe it best suited to follow-up papers. Additionally, we believe that our current study of building the first optogenetic tool in *B. subtilis*, and the synthetic biology insights we learn along the way, is valuable to multiple research communities and warrants its own paper.

The authors purport that their findings generally apply to the genetic manipulation of *B. subtilis* but this claim appears somewhat unsubstantiated given that a single genetic system was studied.

Our results are consistent across our current set of experiments (codon optimization of ORF initial sequences increased expression in 5 genes, terminators increase expression from both the *amyE* and *thrC* loci, etc). However, the reviewer raises a good point that a more thorough validation of each would be necessary to claim generality. That said, we believe our current observations can serve to motivate future larger scale studies. Thus, we have added a few

sentences in the discussion section to clarify that these results are not general, and that further studies need to be conducted.

Parts of the manuscript are hard to grasp on first reading because of the technical lingo and many abbreviations used, some of which are only introduced much later in the manuscript (eg MEFL, MEAP, APC).

We thank the reviewer for pointing this out. We have now introduced all acronyms and abbreviations.

Reviewer #2

Phototoxicity effects are not currently quantified in the manuscript. Comparing the light intensity curve (Fig. 6B) with that of the version 3 CcaSR in *E. coli*, the maximum steady state output is reached with an order of magnitude greater light intensity. It would be helpful to include data supporting the claim in line 294 that phototoxic effects are not a concern.

We have conducted an additional experiment on final OD600 of cultures grown under red or green light, and further analysis of the flow cytometry data shown in Supplementary Figures 16, 17, and 19. These results are now shown in Supplementary Figure S21. We found that, while the density of *B. subtilis* CcaSR v1.0 cultures grown under green light for 12 hours is around 50% lower, this effect disappears once IPTG and xylose are not present (i.e. CcaS and CcaR are not expressed). Thus, the green light is not toxic. Furthermore, this effect seems to be correlated with sfGFP fold-change across many IPTG and xylose induction levels. In addition, this effect is absent in CcaSR v0.2 and CcaSR v0.2- Δ PCB, where output sfGFP expression is lower. These data suggest that maximally induced sfGFP expression from CcaSR v1.0 is so high that it has a deleterious effect on growth – similar to any protein overexpression system. If it were a problem for a future application, this growth defect could easily be reduced by weakening the ribosome binding site in front of the CcaSR output gene.

Regarding the PcpcG2-rspD vs PcpcG2-veg (Fig. 4 vs Fig. S12): Why is the veg variant superior? It seems that the rspD has a higher fold change and lower leakage. What is the rationale for selecting veg?

We thank the reviewer for pointing this out. We have performed additional statistical analysis on the fold-change values from these promoters in response to IPTG, and determined that they are not statistically different ($p \sim 0.1$, two-sample T-test). Furthermore, we think that expression from a strong promoter can easily be brought down, if necessary, using RBS tuning techniques such as the introduction of short sequence repeats between the RBS Shine-Dalgarno region and the ORF (Egbert and Klavins, PNAS 2012). Thus, it is desirable to select the strongest promoter, in this case $P_{cpcG2-veg}$. We have updated the TOM optimization section to better describe this rationale.

CcaSm3 performs better in this study, but ccaSm10 is superior in the version 3 paper in *E. coli*. Please comment on any potential reason for this difference. (Fig. 5).

After performing some additional statistical analysis on the dynamic range values with ccaSm3 and ccaSm10 (two-sample T-test), we have determined that these are statistically identical. Furthermore, we noticed that both in *E. coli* and *B. subtilis*, ccaSm10 results in lower output expression levels compared to ccaSm3. The reason why this results in a higher dynamic range in *E. coli* but not in *B. subtilis* is unclear. We have added a few sentences in the “CcaS miniaturization increases dynamic range” section describing these results.

In Fig. S5 the authors claim that the added terminator accounts for most of the increase in expression. However, they lack a construct that has no terminator and codon optimized pcyA**. This point is relevant in the context of providing information for future *B. subtilis* synthetic biology efforts. It would be useful to know what the impact is on full codon optimization in isolation of the terminator effects.

We thank the reviewer for this suggestion. We have constructed a strain where fully-codon optimized pcyA** is expressed without a terminator downstream. This strain does not result in high fluorescence levels, however, confirming the large effect that the terminator has in gene expression from this locus. These results can now be found in the updated Supplementary Figure 5. Furthermore, the suggested control eliminates the need to describe the partially optimized pcyA*. Thus, we have decided to remove every reference to it.

Fig. S20: sfgfp* (15 bp codon optimized) does show higher expression (B). However, the unpaired probability seems to be lower than that of the regular sfgfp (C vs D). This seems potentially inconsistent with the explanation about why codon optimization increases expression. Please clarify.

We believe an increase in expression still occurs because secondary structure within the ORF region (as opposed to between the ORF and the RBS) is reduced, as indicated by an increase in the unpaired probability of the first 15 bases of sfgfp*. This is consistent with a recent study where secondary structure in the initial ORF sequence was found to affect translation rates by up to 530-fold (Espah Borujeni *et al. Nucleic Acids Res.* 2017). We have added this potential explanation to the figure legend and to the relevant paragraph in the discussion section.

Line 501: These are three first order differential equations, not third order differential equations.

Thank you for pointing this out. We have removed the erroneous “third order” label from this section of the text.

Fig. 2 caption typo: “absence of presence”

We have fixed this typo.

Fig. 4: In this setup, how would CcaR be phosphorylated to act on the PcpG2 variants (as these are in the absence of CcaS, if I understand correctly)?

We thank the reviewer for pointing out this omission. The most likely explanation is a low level of phosphorylation from a small molecule donor such as acetyl-phosphate, as it occurs with many response regulators non-specifically (McCleary, Stock. *J. Biol. Chem.* 1994). We have added a sentence explaining this mechanism to the figure 4 legend.

Line 112: This is a very minor suggestion to improve readability, but it took me a minute to figure out what the “M” was in PPM, etc. so I might suggest moving this to read as “PCB production module”, etc.

Thank you. We have incorporated in the relevant section where the modules are first introduced.

Line 222: sfgfp* is introduced in the text, but it isn't clear what this is. I did eventually find it in the figure caption.

We have modified the text so that sfgfp* is defined the first time it is described.

Line 395: please update “for a few hours” to use more precise language.

We grow cultures inoculated from colonies until turbidity is visible, which takes approximately two hours. We have updated our description of the protocol accordingly.

Methods: inconsistent capitalization on rpm vs. RPM.

We have fixed this inconsistency throughout the Methods section of the manuscript.

Reviewers' Comments:

Reviewer #2:

Remarks to the Author:

The authors have generally addressed my concerns with the revised version of the manuscript. I have one minor suggestion for clarification, described below.

Regarding the new experiments showing that phototoxicity is not an issue:

It would be helpful to provide clarifying text in the caption to avoid confusion about the issue described below. The authors chose to measure cell density in CcaSR v0.2 and CcaSR v0.2- Δ PCB, which intuitively one might think consists of PPM v0.2, LSM v0.2, and TOM v0.2. However, Fig. 2 shows that PPM v0.2 does not produce PCB so the Δ PCB would not be relevant. After looking more closely, CcaSR v0.2 actually consists of the following: PPM v0.3, LSM v0.2, and TOM v0.2 (Fig. S16). So the experiment is good, but the naming and clarity of what exactly was used may be confusing.

Author responses to reviewer comments

Reviewer #2

The authors have generally addressed my concerns with the revised version of the manuscript. I have one minor suggestion for clarification, described below. Regarding the new experiments showing that phototoxicity is not an issue: It would be helpful to provide clarifying text in the caption to avoid confusion about the issue described below. The authors chose to measure cell density in CcaSR v0.2 and CcaSR v0.2- Δ PCB, which intuitively one might think consists of PPM v0.2, LSM v0.2, and TOM v0.2. However, Fig. 2 shows that PPM v0.2 does not produce PCB so the Δ PCB would not be relevant. After looking more closely, CcaSR v0.2 actually consists of the following: PPM v0.3, LSM v0.2, and TOM v0.2 (Fig. S16). So the experiment is good, but the naming and clarity of what exactly was used may be confusing.

Thanks to the reviewer for pointing this out. We have modified the caption of Supplementary Figure 21 so that the modules that compose CcaSR v0.2 and CcaSR v0.2- Δ PCB are explicitly listed.